# Prognostic Significance of aVR Lead and QTc Prolongation in Patients with Early Repolarization

**DOI:** 10.3390/medicina61081466

**Published:** 2025-08-14

**Authors:** Ertugrul Altinbilek, Abuzer Coskun, Burak Demirci, Ismail Oymak, Mustafa Calik, Derya Öztürk, Mustafa Ahmet Afacan, Burcu Bayramoglu

**Affiliations:** 1Department of Emergency Medicine, Istanbul Sisli Etfal Training and Research Hospital, 34371 Istanbul, Turkey; drderyaozturk@yahoo.com; 2Department of Emergency Medicine, Istanbul Bagcilar Training and Research Hospital, 34200 Istanbul, Turkey; dr.acoskun44@hotmail.com (A.C.); drburakdemirci@hotmail.com (B.D.); drismailoymak@gmail.com (I.O.); 3Department of Emergency Medicine, Istanbul Gaziosmanpasa Training and Research Hospital, 34245 Istanbul, Turkey; drmustafacalik@yahoo.com; 4Department of Emergency Medicine, Istanbul Haydarpasa Numune Training and Research Hospital, 34668 Istanbul, Turkey; drmustafaahmet@hotmail.com; 5Department of Emergency Medicine, Istanbul Sancaktepe Sehit Prof. Dr. İlhan Varank Training and Research Hospital, 34785 Istanbul, Turkey; drburcubayramoglu@gmail.com

**Keywords:** emergency department, early repolarization, QTc interval, aVR, acute coronary syndrome

## Abstract

*Background and Objectives*: Early repolarization (ER), previously considered benign for many years, is now recognized as a substantial risk factor for malignant arrhythmia, coronary artery disease, and mortality. The ER pattern, previously regarded as a benign electrocardiographic characteristic, has lately been demonstrated to have a strong association with malignant arrhythmias, coronary artery disease, and elevated death rates. This study seeks to illustrate the prognostic significance of QT interval (QTc) prolongation on electrocardiograms for acute coronary syndrome and death in emergency cases. Prolongation of QTc duration on electrocardiography in emergency room cases may serve as a possible predictor for acute coronary syndrome and mortality. *Materials and Methods*: A retrospective study was conducted on 924 patients diagnosed with ER in the emergency department from 2020 to 2023. The QTc durations, serum glucose levels, troponin I levels, and additional auxiliary data of the patients were assessed. The alteration in the aVR derivation, often overlooked and deemed insignificant, was compared with clinical severity in ER cases. *Results*: The average age of the 924 patients in the study was 48.43 (9.65) years, with 30.7% being female. In the non-cardiac group, the mean age was 51.67 (7.98) years, with 738 females (31.9%). The QTc interval in the patient group was 409.54 (33.46) ms, while in the control group it was 411.57 (27.91) ms (*p* < 0.001). The one-year death rate was 49 (5.3%) for the ER group and nine (0.9%) for the follow-up group. The most common comorbid condition in both groups was tobacco and/or tobacco product intake. Univariate and multivariate regression analyses conducted for both ER types and mortality indicated that QTc prolongation may serve as a predictive factor (*p* < 0.001). The sensitivity and specificity of prolonged QTc in predicting acute coronary syndrome and mortality were determined to be 76% at the lowest and 98% at the highest levels in ER cases (*p* < 0.001). The Kaplan–Meier survival analysis for ER types revealed 321 days for the horizontal type and 363 days for the ascending type. *Conclusions*: Prolonged QTc duration in early repolarization cases may serve as an independent predictor of acute coronary syndrome and mortality.

## 1. Introduction

Early repolarization (ER) is classified as the elevation of the QRS–ST junction or J point on a 12-lead electrocardiogram (ECG) [1]. The ECG interpretation recommendations established in 2009 by the American Heart Association, Heart Rhythm Society, and American College of Cardiology defined ER as J-point elevation with an upsloping or normal ST segment due to the numerous descriptions of ER features [2]. ER is typically observed in individuals who do not have cardiovascular disease [3]. ER is a condition found in approximately 1% to 13% of adults, a pattern that is particularly common in young athletic men [4], and in individuals aged 35–54 years, the presence of ER is associated with a 2- to 4-fold increased cardiovascular mortality [5].

ER morphology is categorized into two types: ascending and horizontal. ER ascending (benign) morphology is defined by a 0.1 mV elevation in the ST segment occurring within 100 ms post-J point, along with a gradual integration of the ST segment into the T wave. In horizontal (malignant) morphology, a 0.1 mV ST segment elevation is observed within 100 ms following the J point and remains flat until the onset of the T wave [6]. Tikkanen et al. [7] demonstrated that the horizontal form is linked to a higher risk of sudden cardiac death in the general population when compared to the ascending form. Uberoi et al. [8] indicated that an ascending ST segment does not correlate with mortality in the emergency room. Recent evidence indicates that the early repolarization (ER) phenomenon represents a spectrum of analogous molecular, ionic, and cellular mechanisms associated with the “J wave” [9].

J waves are related to Ito-mediated amplification, which may trigger life-threatening ventricular arrhythmias [10]. The mechanism underlying J wave formation during acute ischemia remains ambiguous. Evidence indicates that abnormalities in repolarization or depolarization are fundamental to this phenomenon. The first perspective indicates that J wave formation is associated with a direct, transmural variation in the intensity of the outward current [11]. Nonetheless, an alternative mechanism may function under ischemic conditions marked by conduction delay in the impacted areas. It is proposed that an alternative mechanism, defined by conduction delay in the affected areas under ischemic conditions, may also contribute to the phenomenon. Another view is that J wave amplitude increases in tachycardia and that this is related to the conduction delay of the ER pattern [12]. In addition, a decrease in sodium current may also cause J wave development [13].

The prognostic significance of ER in acute coronary syndromes (ACS) has been limited to a limited number of patients, short-term follow-up, and a few case–control studies [14]. The basic mechanisms responsible for ST elevation and ventricular fibrillation in the early stages of ACS and J-wave syndrome are thought to be similar. In addition, ER in the inferior leads has been reported to be associated with an increased risk of fatal ventricular arrhythmias in patients diagnosed with ACS [15]. In rare studies investigating the association of ACS with ER, it has been suggested that ER may increase the risk of ventricular tachycardia and fibrillation [16]. Augmented voltage right (aVR), whether elevation or depression alone, is an important cause of mortality. If ER accompanies a change in aVR lead, it is thought that the clinical prognosis may be worse. aVR lead is located opposite the lateral wall of the apex of the left ventricular cavity. It plays an significant role in the evaluation of electrical activity in this region [17].

The American Heart Association (AHA) called attention to the fact that ST segment elevation in lead aVR in 2013 suggested that it could provide important prognostic information in certain clinical situations [18]. Lead aVR ST segment elevation may indicate multivessel ischemia and left coronary artery blockage, according to the 2017 European Society of Cardiology (ESC) [19]. Publication of the Fourth Universal Myocardial Infarction Definition in 2018 acknowledged ST-elevation myocardial infarction as being identical to ST elevation in lead aVR with particular repolarization patterns [20]. A prolonged QT interval (QTc) is a significant risk factor for polymorphic ventricular tachycardia. QTc has been shown to be associated with high arrhythmia and sudden death in the normal population and in patients with ACS [21]. However, data on early QTc changes in ACS and ER and its prognostic significance are limited [22].

According to the evidence collected until now, it is still unclear if ER is an indication of ACS or arrhythmia. The presence of ER with ACS may provide a novel perspective for assessing the link between aVR lead change and QTc duration in patients with concomitant illness. Controlling or eliminating risk factors is a significant step toward minimizing cardiovascular disease. As a result, in cases diagnosed with ER during admission to the emergency department, the goal was to determine whether there is an association between ER and ACS, as well as to investigate the long-term cardiovascular consequences of ST segment elevation and depression in aVR lead and QTc duration prolongation in terms of morbidity and mortality.

## 2. Materials and Methods

### 2.1. Study Design and Population

The study was conducted with cardiac or non-cardiac patients who were admitted to Bagcilar Training and Research Hospital Emergency Medicine Clinic, Istanbul/Turkey, between 1 January 2020 and 31 December 2023, and whose ECGs showed early repolarization. This retrospective study included 924 patients aged 18 years and older who presented to the emergency department. The hospital automation system encompasses patient diagnoses, hospitalization dates, contact information, demographic details, clinical data, and laboratory results. A total of 3238 individuals participated in the study, including 924 patients diagnosed with early repolarization and a control group of 2314 individuals matched for average age and comorbidities, but lacking early repolarization findings. The patient group and the control group were monitored for one year using hospital automation and e-nabiz systems. Consequently, an evaluation of the ACS and mortality status for both the ER group and the control group was conducted over a one-year period. Patients with acute coronary syndrome were monitored through emergency room and/or cardiology outpatient clinic visits, hospital automation systems, patient registration data, and our country’s e-nabiz system. These patients were monitored for changes in comorbidity, laboratory data included in the study, ICD, and ECGs. Figure 1 presents the distribution of the cases.

### 2.2. Data Collection

Cases diagnosed with ER in the emergency department were categorized into two groups based on the morphology of the J wave, labeled as “horizontal” and “ascending.” Additionally, they were classified into three groups according to the derivation of ER on the electrocardiogram: inferior (II, III, and aVF), inferolateral (II, III, aVF, V5-6), and anterior (V1-6) [23]. During patient follow-up, three groups were identified based on ACS status: individuals who did not develop ACS, those with ST-elevation myocardial infarction (STEMI), and those with non-ST-elevation myocardial infarction (NSTEMI). During follow-up, the status of the aVR derivation observed on the ECG led to the establishment of three groups: normal aVR, aVR ST elevation, and aVR ST depression. The comorbidity and mortality status of the cases was documented. Data on age, gender, glucose levels, lipid levels, and cardiac troponin I were extracted from the emergency department records and patient files of the subjects. Furthermore, ECG findings and QTc intervals were assessed and documented using the automation system. The re-applications of patients were reviewed using the automation system over an average follow-up period of one year. The case results were derived from the national e-nabiz system and the hospital’s automation system, with telephone communication utilized in exceptional circumstances when required. All cases, whether included or excluded from the study, necessitated the absence of COVID-19 infection or a negative polymerase chain reaction test result. This approach facilitated the assessment of a more uniform patient population.

### 2.3. Inclusion and Exclusion Criteria

Patients hospitalized to our emergency medicine clinic with an ER diagnosis between 2020 and 2023 participated in this study. Patients who met all trial requirements were evaluated by two impartial emergency medicine professionals. The electrocardiograms (ECGs) taken at admission were blindly evaluated by a group of specialists. We used a printed ruler in addition to the monitoring equipment to manually take all of our measurements. When there were differences in interpretation, the assessors discussed it and eventually came to a conclusion.ST depression is defined as a maximum depression of 0.1 mm, while ST elevation is defined as a minimum elevation of 0.1 mm in at least two adjacent leads. The research enrolled everyone who was 18 years old or older and fulfilled these requirements. Patients whose blood glucose levels were not assessed during the first 24 h, those with missing or incomplete registration data or laboratory findings, and those less than 18 years old were all deemed ineligible. Conditions related to the central nervous system, the brain, the liver, the kidneys, the inflammation, the cancer, and the blood were not considered in the research. Because there are a lot of things that can cause the QTc interval to be longer, the study also did not include people who had symptoms of hypokalemia, hypocalcemia, hypomagnesemia, postcardiac arrest, high intracranial pressure, congenital long QT syndrome, or were taking specific medications. While both aVR leads and QTc were calculated, patients with QRS width, right bundle branch block, left bundle branch block, and left ventricular hypertrophy were excluded from the study.

### 2.4. Definitions

*2.4a-Acute Coronary Syndrome*: Ia-A patient presenting with sudden chest pain should first have an ECG and troponin I testing. The following ECG criteria for ST-elevation myocardial infarction (STEMI) have been developed by committees of the American College of Cardiology (ACC), the European Society of Cardiology (ESC),the American Heart Association (AHA), and the World Heart Federation (WHF).With the exception of leads V2 and V3, all of the leads show ST segment elevation at the J point, exceeding the threshold of 0.1 mV. Leads V2 and V3 have a threshold of more than 0.2 mV for males older than 40 years, 0.25 mV for males younger than 40 years, and 0.15 mV for women [20]. Ib-NSTEMI, or non-ST elevation myocardial infarction, is defined using myocardial damage indicators, such as troponin I, troponin T, or the MB isoenzyme of creatine phosphokinase. These factors show changed levels in this situation, which is linked to severe ischemia and myocardial damage. Detection of a marker for myocardial injury characterizes non-ST elevation myocardial infarction [18].

*2.4b-aVR ST segment change*: While previous research has shown varying degrees of ST segment elevation and depression in the aVR lead, our study used Wong et al. [24] as its citation. If the aVR lead was more than 0.1 mm elevated or depressed, it was considered an abnormality.

*2.4c-Hypertension*: Hypertension is defined as blood pressure readings of 140 over 90 mm Hg or greater taken more than twice during hospitalization or medication treatment for hypertension. IV- Diabetes mellitus is defined as a fasting blood glucose level of 126 mg/dL or higher or the receipt of antidiabetic treatment.

*2.4d-e-nabiz:* It is a health data recording system that is established by the Ministry of Health in our country, specific to patients and monitored with a special password after ethical approval is obtained, provided that all data remain confidential. This system allows patients to monitor all their data, including previous different hospital applications, polyclinics, laboratory, and imaging.

### 2.5. Design of the Laboratory

Patients were asked to provide biochemical blood samples and serum cardiac troponin I levels when they were admitted to the emergency room. We used the Beckman Coulter Automated AU-680 (Beckman Coulter, Inc., Fullerton, CA, USA) to examine the biochemistry of the blood. We studied the biochemical data within45 to 60 min. Troponin I levels were measured using Troponin I STATE lecsys and Cobas e411 Hitachi (Roche, Basel, Switzerland) analyzers. Results that are above the standard range of 0–0.05 ng/mL for troponin I are considered significant.

### 2.6. Using ECG and QT Interval Computation

An electrocardiography-9132K (12 channels) from Nihon Kohden in Tokyo, Japan, was used to record electrocardiograms (ECGs) taken by patients while they were unconscious. A typical 12-channel electrocardiogram (ECG) device with a paper speed of 25 mm/s and a calibration of 1 mV/10 mm was used to record the patients’ heart rates as soon as they were admitted to the emergency department. They were then entered into the automated system after recording. The duration of the QRS complex was defined as the time it took for the T wave to end, and this was used to measure the QTc. Exclusion criteria for the study were a low frequency of encounters with patients whose electrocardiogram QRS widths were more than 0.12 s. We used the highest QT interval reported in any derivation after evaluating the QT interval in numerous derivations. The second derivation was the most common, although the V5 derivation was also used when observations like obviously flat T waves were not available. Two seasoned specialist doctors used five consecutive beats in the second derivation to assess QTc duration and RR intervals. A combination of automated technology and manual procedures were used to conduct the measurement. Inter-observer agreement was assessed. Data were added when two observers manually and other observers automatically agreed on all measurements. When there was a discrepancy, the average of the two numbers was calculated. The research included little U waves (less than 25% of the T wave) if they occurred with T waves or bifid T waves, but it did not include isolated U waves. To make sure the QTc duration was constant regardless of the heart rate, we employed heart rates ranging from 60 to 100 beats/min. No U waves were detected since the study did not include patients who had electrolyte imbalances. Evaluation specialists disregarded computed QTc durations due to a small sample size of individuals with tachycardia and bradycardia, which would have affected the results. After adjusting for heart rate, the QTc duration was determined. Most commonly used to determine QTc duration is the Bazett formula (QTc = QT/RR1/2) [25]. The normal duration of a QTc interval is 360 ms or longer; a duration of more than 450 ms in males and 460 ms in women is considered protracted [26].

### 2.7. Ethical Considerations

With decision number 2024/03/09/033, the Non-Interventional Clinical Research Ethics Committee of the Health Sciences University Bagcilar Training and Research Hospital gave their approval to this study on 22 March 2024. All participants’ contact information, personal details, and identifying data will remain confidential. At all points, this study followed the guidelines laid out in the Helsinki Declaration for Research Projects.

### 2.8. Statistical Analysis

#### 2.8.1. Power Analysis

During the specified study period, ECGs of 21,759 non-cardiac cases were scanned in the emergency department. In the analysis using G*Power version 3.1.9.7 [27], 924 emergency patients were selected from a population of 2394 cases that met the acceptance criteria with a 5% acceptable margin of error and a 95% confidence interval. Additionally, 2314 non-cardiac and non-ER cases who applied to the hospital were followed up for control purposes. The purpose of keeping the control group large was to see the distribution more clearly and to ensure reliability. The average age, gender distribution, and comorbidity status of the follow-up patients were selected to be close to the patient group. Thus, it was aimed to determine the homogeneous development of the ER and non-ER groups in one year. As a result, the cardiac and mortality statuses of the patient and control groups were determined by automation and the e-nabiz system.

#### 2.8.2. Statistics

The statistical package SPSS 26.0 (SPSS Inc., Chicago, IL, USA) was used to analyze the study’s data. The variables’ normality was checked using the one-sample Kolmogorov–Smirnov test. As none of the variables were normally distributed, we utilized the Kruskal–Wallis H Test to compare the groups based on early repolarization ECG lead, ACS, and aVR alterations, and the Mann–Whitney U Test to compare the groups based on ER morphology and mortality. The associations between groups of nominal variables were investigated using chi-square analysis. Mortality, ACS, aVR alterations, and endovascular morphology were all examined using Spearman’s Rho correlation chart. For the survival status of emergency room cases at one-year follow-up, Kaplan–Meier analysis was computed. In addition, binary logistic regression was used for both univariate and multivariate factor analysis. In order to find independent characteristics that could predict patient groups and mortality outcomes, the variables that were statistically significant in the univariate regression analysis were added to the multivariate regression model using the forward stepwise technique. In binary logistic regression analysis, the association between variables was shown to be statistically significant at *p* < 0.01. A correlation between variables was considered random when the value was higher than *p* > 0.01 and insignificant when the value was higher than *p* > 0.05. Changes in horizontal, ascending, and mortality ER morphology with respect to QTc duration were tested for sensitivity and specificity using receiver operating characteristic (ROC) curve analysis. Results for binary logistic regression were considered statistically significant with a *p*-value less than 0.01 and other variables with a *p*-value less than 0.05.

## 3. Results

The age distribution of the 924 patients included in the study was 25–69 years, with 284 (30.7%) being female. The mean age was 48.43 (9.65) years. The average age of the non-cardiac follow-up group was 51.67 years (SD = 7.98), with 738 participants (31.9%) identifying as female. The analysis revealed a statistically significant difference in gender distribution between the groups (*p* = 0.001). QTc interval was 434.05 (37.74) ms in the horizontal ER group, 398.67 (26.41) ms in the ascending ER group and 411.57 (27.91) ms in the control group (*p* < 0.001). In addition, the mean troponin I value of the ER group was 0.08 (0.19) pg/dL and 0.05 (0.09) pg/dL in the LICHEN follow-up group (*p* = 0.003). Apart from this, glucose was 112.7 (34.49) mg/dL in the ER group and 114.83 (29.72) mg/dL in the follow-up group (*p* < 0.001). Mortality was seen in 49 (5.3%) cases in the ER group and in 9 (0.9%) cases in the non-cardiac follow-up group during the one-year follow-up. When comorbidities were examined, diabetes mellitus was present in 132 (14.3%) and 287 (12.4%) patients, hypertension in 106 (11.5%) and 319 (13.8%) patients and tobacco use in 244 (26.4%) and 502 (21.7%) patients in the ER and follow-up groups, respectively (Table 1). In addition, no complications were observed in 2248 cases in the non-cardiac follow-up group, while STEMI was detected in 34 (1.5%) cases and NSTEMI in 23 (1%) cases. Data are described in Figure 1.

The inferior type was seen most frequently in the early repolarization horizontal group, while the inferolateral type was more common in the ascending group. However, inferior type was found more frequently in total (*p* = 0.048). In addition, mortality was significantly higher in the inferior type (*p* = 0.002). All cases with acute coronary syndrome were more common in the horizontal type (*p* = 0.001). Mortality was also seen most frequently in STEMI patients (*p* = 0.001). ST elevation in aVR derivation was also more common in the horizontal type with 69 cases. In addition, mortality was much higher in aVR ST elevation compared to the depression group (Table 2, *p* = 0.001).

In the correlation analysis between early repolarization (horizontal-ascending) and age, QTc duration, cardiac troponin I and blood glucose levels, a weak to moderate negative correlation was detected for the variables. However, in the analysis of the relationships between aVR derivation, acute coronary syndrome and mortality, a weak correlation was observed with cardiac troponin and a moderate positive correlation was observed with age, QTc duration and blood glucose levels (Table 3).

In the univariate and multivariate analysis of early repolarization (horizontal-ascending), age, QTc duration, troponin I, and glucose level, it was determined that all parameters could be predictive values. In addition, it was seen that other parameters except ST segment change in aVR derivation and blood glucose level could be predictive values for mortality (Table 4).

The receiver operating characteristic (ROC) curve and Kaplan–Meier survival analysis of mortality with QTc duration and early repolarization (horizontal-ascending) are provided in Table 5. The analysis of early repolarization horizontal ROC curves is presented in Figure 2, with Figure 2a depicting the horizontal type and Figure 2b illustrating the ascending type. Figure 3 also shows mortality, while Figure 4 shows the Kaplan–Meier survival analysis.

## 4. Discussion

There are many studies on early repolarization. However, the number of studies showing late-term effects is low. In fact, almost all studies on late-term effects of ER are related to malignant arrhythmias. However, as far as we could scan the literature on QTc duration, acute coronary syndrome, and its change in aVR leads, there are only case-level studies. This situation led us to investigate the relationship between early repolarization types and QTc, ST elevation-depression in aVR lead, and acute coronary. We found that the rates of acute coronary syndrome increased approximately 2-fold in one-year follow-up of cases diagnosed with early repolarization due to non-cardiac reasons in the emergency department, independent of existing risk factors, compared to the normal population. In addition, the change in aVR lead, which is not taken into account by many authors but is considered equivalent to myocardial infarction alone, was detected in 10.8% of cases with early repolarization. It was determined that both ST segment changes in aVR derivation and ER and risk factors increased both morbidity and mortality by 5.44 times compared to the normal population. In addition to the risk factors that can be seen in the normal population, these increases may be important for new conditions that may occur with the contributions of ER. This study contributes to the literature by demonstrating that while ER patterns affect all-cause or cardiovascular mortality by 5.44 times, they are strongly associated with an increased risk in patients with a history of acute coronary syndrome, particularly the aVR or horizontal form. Furthermore, diagnostic confirmation, initiated not only in cardiology clinics but also in the emergency department, will be an important guide to reducing this risk. These findings highlight the importance of considering ER patterns in clinical risk assessments, particularly in high-risk populations. The study is remarkable in that it is one of the rare studies conducted to evaluate the effects of QTc duration and aVR derivation changes on one-year mortality rates and developing comorbidities within the scope of ER cases.

***Early repolarization formation mechanism and frequency:*** Early repolarization, which can be seen in approximately 1–13% of the general population, has long been associated with benign outcomes [4,28]. However, a relationship has been described between early repolarization observed in inferolateral leads and sudden cardiac death. There may be two possible reasons for the difference observed in the incidence of early repolarization. First, the definition and interpretation of ER used in different studies vary. Second, there are significant differences in the basic characteristics of the studied populations. It is noteworthy that 75% of all studied patients showing early repolarization were male. This suggests that gender may be an effective factor in the development of ER [29]. J-point elevation is more common in patients with idiopathic ventricular fibrillation than in healthy individuals [30]. In parallel, male gender was found to be 75% in cases of sudden cardiac death associated with early repolarization [4,7].

The most widely supported hypothesis for the mechanism of early repolarization states that the J-point elevation is due to dissipation of repolarization. Ito and IKATP potassium channels exhibit greater abundance in the epicardium compared to the endocardium. Mutations in genetic channels, vagotonia, and hormonal alterations may lead to a situation where the outward potassium current mediated by these channels surpasses the inward INa and ICaL currents in the endocardium. This leads to a net outward repolarizing current, resulting in J-point elevation on the surface ECG. The dissipation of repolarization results in phase 2 re-entry arrhythmias. The hypothesis concerning the influence of testosterone on Ito channels elucidates the prevalence of this pattern among men in their early 30s. The prolonged recovery time of Ito channels results in more pronounced J-point changes during bradycardia, leading to a higher incidence of arrhythmias during rest or sleep [31]. Pathogenic mutations are presently identified in only a minority of cases; however, the risk of developing early repolarization pattern is markedly elevated in children of patients diagnosed with early repolarization syndrome [32,33]. The incidence of early repolarization cases in the normal population was 4.3%. In total, 69.3% of cases were male, and the mean age of all cases was 48 years. In addition, 89% of the cases with mortality were male.

***Prognostic importance of QTc in early repolarization cases:*** While different parameters were selected to define ER in many studies, QTc was rarely used. Alterations in QTc reflect both therapeutic interventions and repolarization irregularities due to ion channel dysfunctions [34]. The observed increase in electrical heterogeneity in ventricular myocardial cells may result in an extended QTc duration. This condition is linked to myocardial electrical disturbances caused by dysfunction of the autonomic nervous system, cellular necrosis, and imbalances in electrolytes [21]. Studies indicate that the factors leading to QTc prolongation during acute myocardial ischemia include reduced epicardial temperature, acidosis, alterations in impedance, and electrical heterogeneity within the ventricular myocardium [35]. Ozcan et al. [14] calculated QTc as 395 ms in their 61-case ER study. Although it was higher than the control group, the authors did not calculate ER as horizontal or ascending in this value, which was not significant. Tikkanen et al. [7], who conducted a similar study, calculated the QTc interval as 417 ms and reported that it was longer in horizontal type ER. In our study, QTc was 409 ms in all ER cases, and it was found to be significantly higher in horizontal type than ascending type with 434 ms. In addition, QTc prolongation was significant with ER types, aVR drives, ACS, and mortality in both correlation and regression analysis. In addition, the sensitivity and specificity rate of QTc prolongation with ER groups was determined as 76%, with 97% mortality. We think that QTc prolongation can be a predictive value if it is taken into consideration in ER cases.

The aVR change in the ER population was 10.8%. Lead aVR ST elevation instances accounted for 24.3% of these, whereas ST depression cases accounted for 4.2%. Acute coronary syndrome patients with a lead aVR showing ST elevation ranges from 7.3% to 32.3% [36]. In a study conducted by Misumida et al. [37] on STEMI patients, the authors found that ST elevation in lead aVR was a more reliable diagnostic predictor of left main coronary artery disease than NSTEMI. According to Rathi et al. [38], left major coronary artery disease was more common in the group with aVR-ST elevation as compared to the depression group. A notable rise in mortality associated with elevated lead aVR levels was observed. In our ER cases, 87.8% of the mortality was in the ST elevation group in the aVR derivation. This change in the aVR derivation alone increases the mortality rate by 5.1% compared to the normal population, independently of other risk factors. However, in the literature, Demirci et al. [39] mentioned the aVR derivation changed epidemiologically in their ER study of 442 cases and determined that it was important both in terms of mortality and horizontal type. We believe that the importance of aVR derivation will be better understood in future studies on ER cases.

***Acute coronary syndromes and the QTc interval in cases of early repolarization****:* Kenigsberg et al. [40] showed that the first ischemic event in myocardial ischemia is QTc interval prolongation, which is important for prognosis. Over the course of six years, Mann et al. [41] examined 4936 patients diagnosed with STEMI and discovered a robust and independent correlation between an extended QT and mortality. In addition, QT values can help in identifying patients at high risk and determining how long certain individuals need to be followed up. According to Gadelata et al. [35], a patient’s probability of mortality in NSTEMI is independently predicted by the presence of a QTc interval prolongation on admission electrocardiogram. Prolonged QTc intervals are an important clinical sign for detecting ischemia early on and for identifying individuals at high risk for dangerous arrhythmias [42]. Hamati et al. [3] compared 45 patients who underwent emergency room procedures with 45 patients who did not and discovered that the former group had a higher prevalence of coronary artery stenosis.

Acute ischemia is related with more severe coronary disease when QTc values are high [42]. One useful sign for evaluating the increased risk of acute coronary syndrome may be tracking the fluctuations in QTc over time [26]. Changes in lead aVR are linked to an elevated risk of myocardial infarction in the ACS group. Acute coronary syndrome (ACS) prognosis and mortality can be independently predicted by changes in lead aVR [43]. Taken together, these results highlight the significance of lead aVR alterations and QTc interval extension as risk factors. The risk grows exponentially when the change in lead aVR is coupled with a lengthening of the QTc interval in ACS. Our study’s key conclusions were as follows: (1) A larger ischemic area in an ACS is associated with an increased risk of ischemic heart failure, cardiac tamponade, malignant arrhythmias, and acute pulmonary edema in patients. (2) The QTc distance and distribution are significantly higher in leads related to an ACS. (3) Prolonged QTc distance in conjunction with changes in the aVR lead is a strong predictor of mortality when compared to the general population. (4) We believe that the aVR-QTc association is a crucial parameter for predicting prognosis and mortality.

***Limitations:*** The current study had some limitations. The most important limitation was that it was a retrospective study. Secondly, despite the significant sample size, there were difficulties encountered in follow-up and accessibility. The cases’ left ventricular ejection fraction, chronic medications, and coronary angiography results were not able to be analyzed. If these parameters had been added, it could have been a more effective study. Third, the malignant arrhythmia status of the cause of mortality in the non-cardiac follow-up group and the ER group could not be followed. Fourthly, the QTc values of the cases were calculated at the time of admission to the emergency department, and no re-measurements were made in the subsequent follow-ups. Therefore, the additional burden of this on malignant arrhythmia and mortality could not be compared. Furthermore, the study systematically excluded electrolyte disorders, congenital long QT syndrome, medications that could prolong QTc, and similar factors; however, it is possible that additional factors were overlooked. Because patients presented to the emergency department for non-cardiac reasons, the study did not consider whether heart rate was large enough to justify the use of Fridericia or Framingham corrections. A significant limitation of the study is that all patients with STEMI, NSTEMI, and ST elevation and depression in lead aVR underwent angiography. However, because angiography was not performed on the other non-cardiac patients, generalizations could not be made about the left anterior descending artery (LAD), circumflex artery (Cx), left main coronary artery (LMCA), and right main coronary artery (RCA). Another limitation is that factors such as age, QTc, glucose, and troponin were analyzed in both univariate and multivariate analyses. However, the study may not have accounted for ischemic heart disease history, smoking, medication use, renal dysfunction, or hypertension. These factors may limit their interpretability as independent predictors. In addition, aortic stenosis and valvular heart disease were rare and therefore were not studied. These may be a possible limitation.

## 5. Conclusions

Our study showed that QTc prolongation on ECG in ER cases is an independent prognostic factor affecting the risk of late acute cardiac syndrome and mortality. In addition, serum glucose and troponin I levels were also important predictive parameters. QTc duration, which can be calculated with simple and practical methods, can be a predictive marker that can determine the prognosis for such patients. However, we accept that additional studies are necessary to confirm and compare the prognostic significance of the presence of ER and QTc prolongation.

## Figures and Tables

**Figure 1 medicina-61-01466-f001:**
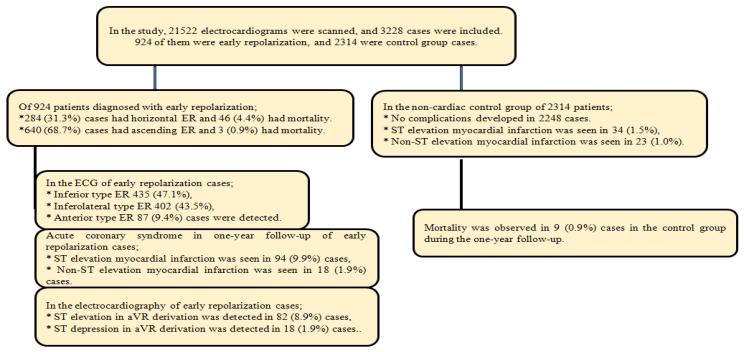
Demographic distribution of cases.

**Figure 2 medicina-61-01466-f002:**
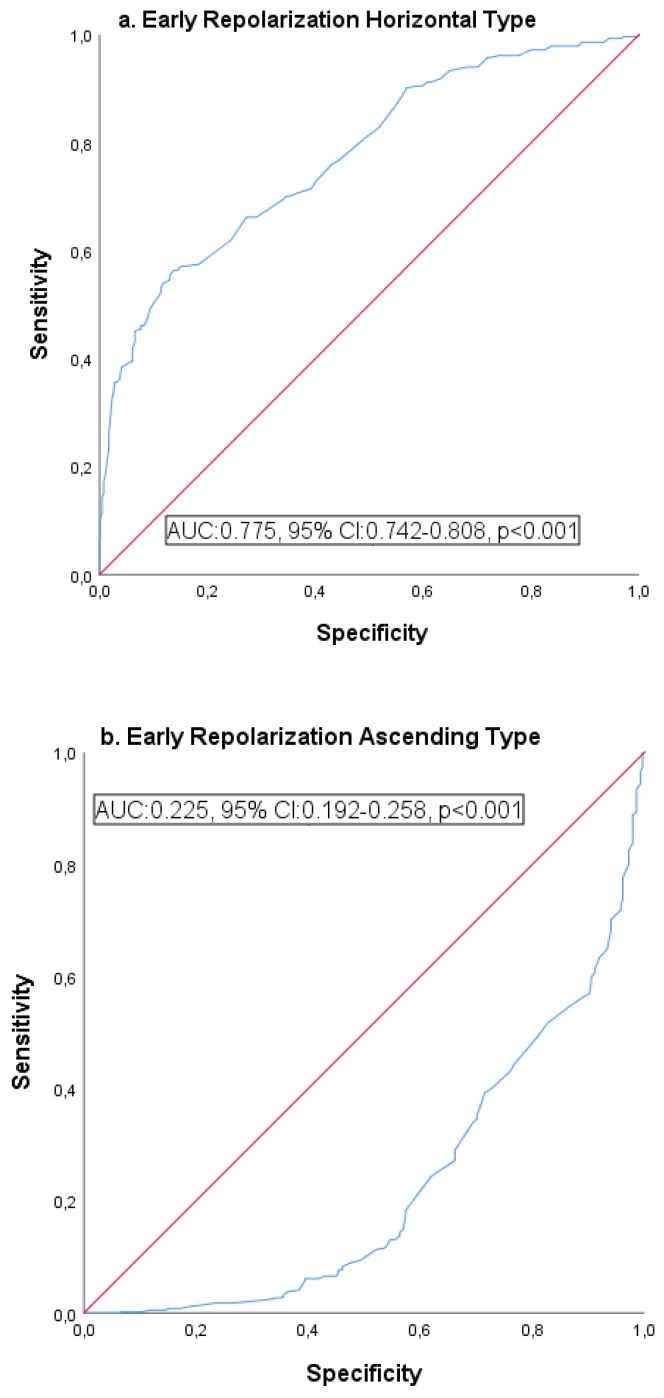
Receiver operating characteristic (ROC) curve analysis: relationship between QTc and early repolarization (horizontal-ascending) types.

**Figure 3 medicina-61-01466-f003:**
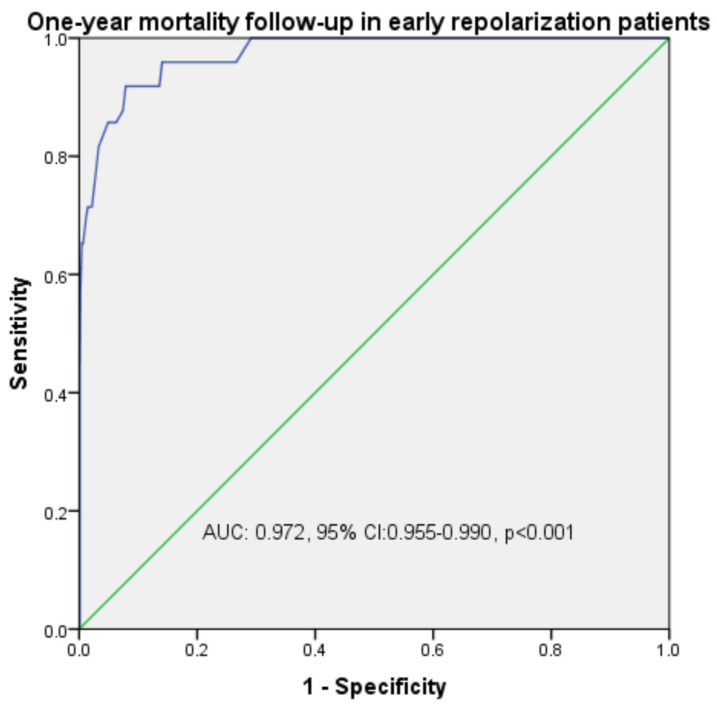
Receiver operating characteristic (ROC) curve analysis:relationship between QTc and mortality.

**Figure 4 medicina-61-01466-f004:**
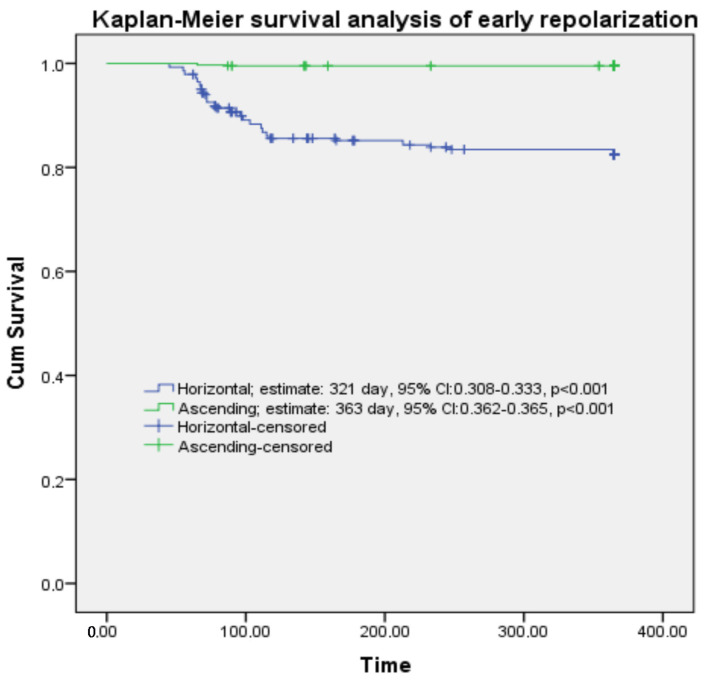
Kaplan–Meier survival analysis of early repolarization horizontal-ascending types.

**Table 1 medicina-61-01466-t001:** Analysis of the basic characteristics and laboratory results of the early repolarization and non-cardiac groups.

Early Repolarization									Non-Cardiac Follow-Up Group	
				All Patient Mean (SD), *n* (%)		Horizontal Mean (SD), *n* (%)	Ascending Mean (SD), *n* (%)		Mean (SD), *n* (%)	*p*-Value
**Age, year**				48.43 (9.65)		53.52 (8.65)	46.16 (9.2)		51.67 (7.98)	**<0.001**
**Gender**		**Female**		284 (30.7)		84 (29.6)	278 (43.4)		738 (31.9)	**0.001**
		**Male**		640 (69.3)		200 (70.4)	362 (56.6)		1576 (68.1)	
**Corrected QT interval,** ms				409.54 (33.46)		434.05 (37.74)	398.67 (26.41)		411.57 (27.91)	**<0.001**
**Time,** day				335.2 (83.27)		279.65 (123.95)	359.85 (35.1)		364.87 (54.38)	**<0.001**
**Lab**	**Troponin I,** pg/dL			0.08 (0.19)		0.13 (0.23)	0.07 (0.16)		0.05 (0.09)	**0.003**
	**Blood sugar,** mg/dL			112.7 (34.49)		127.45 (42.95)	106.16 (27.28)		114.83 (29.72)	**<0.001**
	**Triglyceride,** mg/dL			88.41 (25.93)		90.24 (256.97)	87.5 (25.38)		96.38 (21.67)	**0.005**
	**Cholesterol,** mg/dL			151.84 (41.5)		155.64 (40.12)	149.94 (42.11)		164.31 (44.73)	0.057
	**VLDL,** mg/dL			97.74 (34.26)		99.82 (33.81)	96.7 (34.5)		94.64 (35.92)	0.274
	**HDL,** mg/dL			35.19 (19.11)		36.89 (30.49)	34.33 (9.11)		36.93 (21.85)	0.564
**Mortality**				49 (5.3)		46 (93.9)	3 (6.1)		9 (0.39)	
**Comorbidity**			**Hypertension**		106 (11.5)			319 (13.8)		
			**Diabetes mellitus**		132 (14.3)			287 (12.4)		
			**Tobacco**		244 (26.4)			502 (21.7)		

SD: Standard deviation, VLDL: Very-low-density lipoprotein, HDL: High-density lipoprotein, *p*: Statistical significance (*p* < 0.05).

**Table 2 medicina-61-01466-t002:** Analysis of early repolarization and mortality with variables.

Early Repolarization					Mortality	
		Horizontal *n* (%)	Ascending n (%)	*p*-Value	*n* (%)	*p*-Value
**Early repolarization**	**Inferior**	148 (52.1)	287 (44.8)	**0.048**	35 (71.4)	**0.002**
	**Inferolateral**	113 (39.8)	289 (45.2)		12 (24.5)	
	**Anterior**	23 (8.1)	64 (10)		2 (4.1)	
**Acute coronary syndrome**	**No**	186 (65.5)	628 (98.1)	**0.001**	3 (6.1)	**0.001**
	**STEMI**	86 (30.3)	6 (0.9)		40 (81.6)	
	**NSTEMI**	12 (4.2)	6 (0.9)		6 (12.3)	
**aVR Derivation**	**No**	203 (71.5)	621 (97)	**0.001**	2 (4.1)	**0.001**
	**ST Elevation**	69 (24.3)	13 (2.1)		43 (87.8)	
	**STDepression**	12 (4.2)	6 (0.9)		4 (8.1)	

STEMI: ST-elevation myocardial infarction, NSTEMI: Non-ST-elevation myocardial infarction, *p*: Statistical significance (<0.05).

**Table 3 medicina-61-01466-t003:** Correlation analysis of variables with early repolarization, aVR lead, acute coronary syndrome, and mortality rate.

	Early Repolarization		aVR Derivation		Acute Coronary Syndrome		Mortality	
	r	*p*	r	*p*	r	*p*	r	*p*
**Age**, year	**−0.361**	**<0.001**	**0.426**	**<0.001**	**0.494**	**<0.001**	**0.342**	**0.001**
**Corrected QT interval,** ms	**−0.440**	**<0.001**	**0.489**	**<0.001**	**0.549**	**<0.001**	**0.367**	**0.001**
**Cardiac troponin I,** pg/dL	**−0.096**	**0.003**	**0.099**	**0.003**	**0.110**	**0.001**	**0.141**	**0.001**
**Blood sugar,** mg/dL	**−0.307**	**<0.001**	**0.304**	**<0.001**	**0.357**	**<0.001**	**0.210**	**0.001**

r: Correlation coefficient, *p*: Statistical significance (<0.05).

**Table 4 medicina-61-01466-t004:** Univariate and multivariate analysis of early repolarization, aVR lead, and mortality rate with variables.

Early Repolarization							
		Univariate			Multivariate		
		OR	95% CI	*p*-Value	OR	95% CI	*p*-Value
**Early Repolarization**	**Age**, year	0.910	0.893–0.927	**<0.001**	0.968	0.947–0.989	**0.003**
	**Corrected QT interval,** ms	0.964	0.959–0.969	**<0.001**	0.972	0.965–0.978	**<0.001**
	**Cardiac troponin I,** pg/dL	0.199	0.098–0.401	**0.001**	0.225	0.104–0.489	**0.001**
	**Blood sugar,** mg/dL	0.981	0.976–0.986	**<0.001**	0.991	0.983–0.999	**0.022**
**aVR Derivation**	**Age**, year	1.320	1.249–1.395	**<0.001**	1.091	1.026–1.160	**0.005**
	**Corrected QT interval,** ms	1.083	1.068–1.098	**<0.001**	1071	1.053–1.090	**<0.001**
	**Cardiac troponin I,** pg/dL	8.232	3.705–18.289	**<0.001**	25.410	6.095–105.928	**<0.001**
	**Blood sugar,** mg/dL	1.017	1.011–1.022	**0.001**			
**Mortality**	**Age**, year	1.409	1.298–1.530	**<0.001**	1.230	1.086–1.393	**0.001**
	**Corrected QT interval,** ms	1.124	1.086–1.164	**<0.000**	1.094	1.057–1.133	**<0.001**
	**Cardiac troponin I,** pg/dL	27.800	11.112–69.548	**<0.001**	1784.34	123.32–25818.44	**<0.001**
	**Blood sugar,** mg/dL	1.013	1.008–1.019	**0.001**			

OR: Odds ratio, 95% CI: Confidence interval, *p*: Statistical significance (<0.05).

**Table 5 medicina-61-01466-t005:** Receiver operating characteristic curve and Kaplan–Meier survival analysis of morphology and mortality in early repolarization.

		Early Repolarization				
		AUC	95% CI	*p*	Sensitivity (%)	Specificity (%)
**ROC curve analysis**	**Horizontal**	0.775	0.742–0.808	**<0.001**	83.1	81.7
	**Ascending**	0.225	0.192–0.258	**<0.001**	78.7	76.9
	**Mortality**	0.972	0.955–0.990	**<0.001**	98.4	97.1
**Kaplan–Meier survival analysis**		**Estimate,** day	**95% CI**	** *p* **		
	**Horizontal**	321	308–333	**<0.001**		
	**Ascending**	363	362–364	**<0.001**		

ROC: Receiver operating characteristic, AUC: Area under the curve, 95% CI: Confidence interval, *p*: Statistical significance (<0.05).

## Data Availability

The original contributions presented in this study are included in the article. Further inquiries can be directed to the corresponding author. All data are available on request without restriction.

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
