# Peer review of "Prognostic Significance of aVR Lead and QTc Prolongation in Patients with Early Repolarization"

_medicina, 2025, doi:10.3390/medicina61081466_

Round 1

Reviewer 1 Report

Comments and Suggestions for Authors

Congratulations for such a thorough analysis. In my opinion it is an interesting matter and emphasize once more the importance of standard ECG study which is a simple method extremely important in proper hands. As you noticed in spite to be considered a benign condition, in some circumstances ER with QT prolongation proved to be an important marker of risk. You also recognized the drawbacks of the study, especially the lack of data regarding medication, electrolyte imbalance, comorbidities (thyroid function and blood calcium value) and also the lack of reevaluate the ECG aspects. I don`t think that aortic stenosis might be important. The importance of reinvented this condition and the fact that it deserves a detail research is also the little bit old biographical data regarding the subject. Maybe it is useful for a future project to short follow-up this patients and their comorbidities.

Author Response

        Dear Reviewer,

        Thank you for taking your valuable time. We have carefully read your nice comments and suggestions. As the author team, we have evaluated your opinions and suggestions from point to point. We have made all the edits with your contributions. We have seen that these opinions and suggestions have added value to our article. Thank you again for your valuable comments.

      I would also like to point out that the e-Pulse system used in our country provides significant convenience for our studies. Once a patient enters a hospital, they are entered into the e-Pulse system using the Ministry of Health's data. After receiving approval from the ethics committee, an application is submitted to the Ministry of Health to obtain a password for review. This allows patients included in the study at our hospital to view all their data, even if they visit another hospital. This provides us, the authors, with significant convenience.

  1. Comment: Congratulations on such a thorough analysis. In my opinion, it is an interesting matter and emphasizes once more the importance of standard ECG study, which is a simple method extremely important in the proper hands. As you noticed in despite being considered a benign condition, in some circumstances, ER with QT prolongation proved to be an important marker of risk. You also recognized the drawbacks of the study, especially the lack of data regarding medication, electrolyte imbalance, comorbidities (thyroid function and blood calcium value), and the lack of re-evaluation of the ECG aspects. I don`t think that aortic stenosis might be important. The importance of reinventing this condition and the fact that it deserves detailed research is also the little bit of old biographical data regarding the subject. Maybe it is useful for a future project to follow up on these patients and their comorbidities.

     Your kind comment has encouraged us to work even harder. Thank you very much for that.

  1. Comment: Development of the method section

    Deficiencies identified in the method section were revised. Revised sections are highlighted in green.

    Best regards.

Reviewer 2 Report

Comments and Suggestions for Authors

This manuscript presents a retrospective study investigating the prognostic implications of QTc prolongation and aVR lead changes in patients diagnosed with early repolarization (ER) in the emergency department. The study is timely and clinically relevant, addressing a topic of growing interest, particularly given the evolving understanding of ER as a potentially malignant ECG phenotype. Several areas require revisions to enhance the scientific robustness and clinical impact of the work. Here are my detailed suggestions for improvement:

The study’s strength lies in its attempt to stratify ER morphology (horizontal vs. ascending) and correlate QTc prolongation and aVR changes with ACS and one-year mortality. While the concept is clinically meaningful, its novelty is limited by the observational and retrospective nature of the data and the lack of mechanistic insight. Please clarify how this study adds to existing literature beyond previous ER-related mortality studies. A more focused discussion of how the findings could inform ED triage or follow-up strategies would strengthen clinical relevance.

The method of QTc measurement, although detailed, remains vulnerable to inter-observer variability. Were intra- and inter-rater agreement assessed (e.g., via intraclass correlation coefficients)? If not, this should be acknowledged as a limitation.

The authors used Bazett’s formula, which is known to overcorrect at higher heart rates. Was the heart rate range of the cohort broad enough to justify use of Fridericia or Framingham corrections instead?

The prognostic significance of ST elevation in aVR is well-described in the literature for high-risk coronary lesions (e.g., LMCA). However, the inclusion criteria and rationale for applying this criterion in an ER population with presumed non-cardiac presentations need to be more clearly justified. Please clarify whether angiographic data or cardiac imaging were available for validation of coronary disease severity.

Was there any effort to exclude patients with baseline ECG abnormalities (e.g., LBBB, LVH) that might confound aVR interpretation?

The manuscript claims to track ACS incidence over one year, yet the methods section does not clearly outline how ACS diagnoses were adjudicated. Were all follow-up ACS events confirmed by troponin and ECG criteria or just by hospital ICD coding?

While multivariate analysis was performed, the variables included (age, QTc, glucose, troponin) may not account for all potential confounders (e.g., history of ischemic heart disease, smoking, medication use, renal dysfunction). The absence of adjustment for these factors may limit interpretability of QTc or aVR findings as independent predictors. Consider including a sensitivity analysis or clearly acknowledging the limitation of residual confounding in the discussion.

Author Response

        Dear Reviewer,

        Thank you for taking your valuable time. We have carefully read your nice comments and suggestions. As the author team, we have evaluated your opinions and suggestions from point to point. We have made all the edits with your contributions. We have seen that these opinions and suggestions have added value to our article. Thank you again for your valuable comments.

      I would also like to point out that the e-Pulse system used in our country provides significant convenience for our studies. Once a patient enters a hospital, they are entered into the e-Pulse system using the Ministry of Health's data. After receiving approval from the ethics committee, an application is submitted to the Ministry of Health to obtain a password for review. This allows patients included in the study at our hospital to view all their data, even if they visit another hospital. This provides us, the authors, with significant convenience.

      Comment: This manuscript presents a retrospective study investigating the prognostic implications of QTc prolongation and aVR lead changes in patients diagnosed with early repolarization (ER) in the emergency department. The study is timely and clinically relevant, addressing a topic of growing interest, particularly given the evolving understanding of ER as a potentially malignant ECG phenotype. Several areas require revisions to enhance the scientific robustness and clinical impact of the work. Here are my detailed suggestions for improvement:

           As the author team, we thank you for your valuable comments. 

  1. Question: The study’s strength lies in its attempt to stratify ER morphology (horizontal vs. ascending) and correlate QTc prolongation and aVR changes with ACS and one-year mortality. While the concept is clinically meaningful, its novelty is limited by the observational and retrospective nature of the data and the lack of mechanistic insight. Please clarify how this study adds to existing literature beyond previous ER-related mortality studies. A more focused discussion of how the findings could inform ED triage or follow-up strategies would strengthen clinical relevance.

     Answer-1. Literature contributions for emergency services and cardiology were added to the discussion section.

  1. Question: The method of QTc measurement, although detailed, remains vulnerable to inter-observer variability. Was intra- and inter-rater agreement assessed (e.g., via intraclass correlation coefficients)? If not, this should be acknowledged as a limitation.

    Answer-2. The change you requested has been explained under the QTc measurement heading.

  1. Question: The authors used Bazett’s formula, which is known to overcorrect at higher heart rates. Was the heart rate range of the cohort broad enough to justify the use of Fridericia or Framingham corrections instead?

       Answer-3. A limitation is noted in the study, as we did not examine whether the cohort's heart rate range was wide enough to justify the use of Fridericia or Framingham corrections.

  1. Question: The prognostic significance of ST elevation in aVR is well-described in the literature for high-risk coronary lesions (e.g., LMCA). However, the inclusion criteria and rationale for applying this criterion in an ER population with presumed non-cardiac presentations need to be more clearly justified. Please clarify whether angiographic data or cardiac imaging were available for validation of coronary disease severity.

    Answer-4.  Because angiography was not performed on all patients in the study, simply reporting the items performed was considered to affect the statistical data and was therefore omitted in the limitations section. However, we can still include data on acute coronary syndrome and aVR derivation if desired.

  1. Question: Was there any effort to exclude patients with baseline ECG abnormalities (e.g., LBBB, LVH) that might confound aVR interpretation?

     Answer-5.  When calculating lead aVR and QTc, patients with right bundle branch block, left bundle branch block, left ventricular hypertrophy, and wide QRS were excluded from the study. These were added to the exclusion criteria in the methods section.

  1. Question: The manuscript claims to track ACS incidence over one year, yet the methods section does not clearly outline how ACS diagnoses were adjudicated. Were all follow-up ACS events confirmed by troponin and ECG criteria, or just by hospital ICD coding?

     Answer-6.  How acute coronary syndrome patients were followed up was added to the method section.

  1. Question: While multivariate analysis was performed, the variables included (age, QTc, glucose, troponin) may not account for all potential confounders (e.g., history of ischemic heart disease, smoking, medication use, renal dysfunction). The absence of adjustment for these factors may limit the interpretability of QTc or aVR findings as independent predictors. Consider including a sensitivity analysis or acknowledging the limitation of residual confounding in the discussion.

  Answer-7.   Univariate and multivariate regression analyses were performed on some parameters. However, regression analysis was not performed on comorbidity data. This was added to the limitation section.

    Best regards.

Round 2

Reviewer 2 Report

Comments and Suggestions for Authors

The manuscript has been improved following the revision